# High-Fat Diet Enhances Working Memory in the Y-Maze Test in Male C57BL/6J Mice with Less Anxiety in the Elevated Plus Maze Test

**DOI:** 10.3390/nu12072036

**Published:** 2020-07-09

**Authors:** Kaichi Yoshizaki, Masato Asai, Taichi Hara

**Affiliations:** 1Department of Disease Model, Institute for Developmental Research, Aichi Developmental Disability Center, Aichi 480-0392, Japan; masato-a@inst-hsc.jp; 2Laboratory of Food and Life Science, Faculty of Human Sciences, Waseda University, 2-579-15 Mikajima, Tokorozawa, Saitama 359-1192, Japan

**Keywords:** high-fat diet, psychiatric behavior, cognitive behavior, C57BL/6J, obesity

## Abstract

Obesity is characterized by massive adipose tissue accumulation and is associated with psychiatric disorders and cognitive impairment in human and animal models. However, it is unclear whether high-fat diet (HFD)-induced obesity presents a risk of psychiatric disorders and cognitive impairment. To examine this question, we conducted systematic behavioral analyses in C57BL/6J mice (male, 8-week-old) fed an HFD for 7 weeks. C57BL/6J mice fed an HFD showed significantly increased body weight, hyperlocomotion in the open-field test (OFT) and Y-maze test (YMZT), and impaired sucrose preference in the sucrose consumption test, compared to mice fed a normal diet. Neither body weight nor body weight gain was associated with any of the behavioral traits we examined. Working memory, as assessed by the YMZT, and anxiety-like behavior, as assessed by the elevated plus maze test (EPMT), were significantly correlated with mice fed an HFD, although these behavioral traits did not affect the entire group. These results suggest that HFD-induced obesity does not induce neuropsychiatric symptoms in C57BL/6J mice. Rather, HFD improved working memory in C57BL/6J mice with less anxiety, indicating that an HFD might be beneficial under limited conditions. Correlation analysis of individual traits is a useful tool to determine those conditions.

## 1. Introduction

Obesity is a condition that is characterized by massive adipose tissue accumulation. This condition is generally caused by a complicated interplay between intrinsic and extrinsic variables, including genetic, epigenetic, and developmental factors, as well as low levels of physical activity, high levels of sedentariness, and access to inexpensive and high-calorie foods [1,2,3,4,5]. Obesity presents a great threat to human health by promoting physical inactivity, inflammation, blood lipid disorders, and insulin resistance [6]. Further, certain psychiatric disorders, including major depressive disorder and post-traumatic stress disorder [7,8,9,10,11], as well as neurodegenerative disorders, such as dementia and Alzheimer’s disease [12,13,14], are associated with obesity. Several animal models of obesity have been established via the long-term, excessive intake of a high-fat diet (HFD), which causes significant body weight gain, higher adipose/weight ratio, and higher serum levels of free fatty acids [15,16]. An HFD has been shown to induce anxiety-like behavior, hyperlocomotion, and anhedonia-like behavior [17,18,19], and to exacerbate chronic, unpredictable, mild, stress-induced anhedonia-like behavior [17]. Further, an HFD has shown to have a detrimental effect on operant learning [20,21,22]. Together, these findings indicate that an HFD induces abnormal psychiatric traits and cognitive impairment in rodents.

Dietary therapies, medical treatment, and exercise have been applied as preventative measures and remedies for obesity and metabolic syndrome [23]. Certain pungent spice ingredients such as capsaicin, ginger, and pepper have been shown to inhibit fat accumulation and decrease blood triglyceride levels by increasing energy metabolism [24,25,26]. Furthermore, fatty acid-derived compounds 9- and 13-oxo-octadecadienoic acid, which are found in tomatoes, act as agonists of peroxisome proliferator-activated receptor alpha (PPARα), which regulates fatty acid synthesis in the liver [27]. Moreover, compounds such as auraptene from citrus, isoprenoids from herbs, and β-cryptoxanthin from tangerine and papaya regulate the differentiation and function of adipocytes [28,29]. Thus, many functional foods normalize blood glucose and triglyceride levels and reduce liver fat content. However, the effects of these functional foods have not been examined in relation to HFD-induced abnormal psychiatric traits and cognitive impairment. Therefore, it is unclear whether HDF-induced obesity presents risks for psychiatric disorders and cognitive impairment.

In the present study, we performed systematic behavioral and correlation analyses to examine the relationship between body weight and behavioral traits in male C57BL/6J mice that were fed an HFD. Consistent with previous studies [17,18], we found that the HFD significantly induced body weight gain, hyperlocomotion, and anhedonia-like behavior. However, HFD-induced body weight gain was not correlated with any behavioral traits. In contrast, significant correlation was observed between working memory in the Y-maze test (YMZT) and anxiety-like behavior in the elevated plus maze test (EPMT) in mice fed an HFD. These results suggest that HFD-induced obesity does not promote neuropsychiatric symptoms, such as hyperlocomotion and anhedonia-like behavior in male C57BL/6J mice, but rather that an HFD enhances working memory in C57BL6J mice with less associated anxiety.

## 2. Materials and Methods

### 2.1. Animals

Animal experiment was approved by the Ethics Committee for Animal Experiments of Aichi Institute for Developmental Research (2019-022) and were carried out in accordance with the National Institutes of Health guidance for the care and use of laboratory animals. Seven-week-old male C57BL/6J mice were purchased from CLEA Japan, Inc. (Tokyo, Japan). After a 1-week habituation period, mice were randomly divided into two groups as follows: one was fed a normal diet (ND, *n* = 10, 25.4% protein, 50.3% carbohydrate, 4.6% fat, 3.47 kcal/g, CE-2; CLEA Japan, Japan), and the other was fed an HFD (*n* = 9, 20% protein, 20% carbohydrate, 60% fat, 5.24 kcal/g, D12492; Research Diets Inc., New Brunswick, NJ, USA) for 7 weeks, as described in previous papers [17,19,20]. All animals were maintained under a 12 h dark-light cycle (light from 07:00 to 19:00) at 23 ± 1 °C with *ad libitum* access to food and water.

### 2.2. Open-Field Test

Exploratory activity and anxiety-like behavior were measured using an open-field apparatus (50 × 50 × 50 cm, O′Hara and Co., Ltd., Tokyo, Japan). Each mouse was placed in the center of the open-field apparatus. The center zone was defined as a square, 10 cm away from the wall. Distance traveled and time spent in the center zone by each animal was recorded for 10 min with a video-imaging system (EthoVisionXT; Noldus Information Technology, Wageningen, The Netherlands), as described previously [30].

### 2.3. Elevated Plus Maze Test

Anxiety-like behavior was measured using the elevated plus maze (40 cm length, 10 cm width, 50 cm height; O′Hara and Co., Ltd., Tokyo, Japan). The closed arms were enclosed by a black wall 20 cm in height. Each mouse was placed in the central area of the maze facing one of the open arms. Time spent in the open arms was measured for 10 min with the EthoVisionXT video-imaging system, according to a previous paper [31].

### 2.4. Y-Maze Test

Working memory and exploratory activity were measured using a Y-maze apparatus (arm length: 40 cm, arm bottom width: 3 cm, arm upper width: 13 cm, height of wall: 15 cm, BrainScience Idea, Osaka, Japan). Each mouse was placed in the central area. The number of entries into the arms and alterations were recorded for 10 min with the EthoVisionXT video-imaging system. Working memory was calculated as number of correct alterations/number of total new arm entries, as described in a previous paper [32]

### 2.5. Sucrose Consumption Test

Anhedonia-related behavior was measured using the sucrose consumption test. After an 18-h period of water deprivation, each animal was exposed to two bottles, one containing tap water and the other containing sucrose solution (1%). Water consumption was measured by weighing the bottle before and after the test. Sucrose preference (percentage) was calculated according to the following formula: sucrose solution intake/total intake, where total intake = sucrose solution intake + tap water intake, as previously described [33,34]. The positions of the two bottles were varied randomly.

### 2.6. Statistical Analysis

Data are presented as the mean ± standard error of the mean (SEM). Body weight differences between groups of mice were determined by restricted maximum likelihood estimation followed by post hoc Student’s *t*-test. Differences in behavioral traits were determined by the Student’s *t*-test. Correlations were assessed by Pearson correlation analyses. Prism 8 software (San Diego, CA, USA) was used for statistical analyses, and *p* < 0.05 was considered statistically significant.

## 3. Results

### 3.1. HFD Affects Body Weight Gain, Hyperlocomotion, and Anhedonia-Like Behavior

Body weight was significantly higher in the HFD group than in the ND group after 1 week of treatment (Figure 1a, *t* = 5.087, df = 17, *p* < 0.001). Likewise, body weight gain was significantly higher in the HFD group than in the ND group (Figure 1b, *t* = 4.962, df = 17, *p* < 0.001). Locomotor activity and anxiety-like behaviors were examined using the open field test (OFT), and distance traveled was found to be significantly higher in the HFD group than in the ND group (Figure 1c, *t* = 3.775, df = 17, *p* < 0.01). In contrast, the amount of time spent in the center zone was comparable between both groups (Figure 1d, *t* = 0.463, df = 17, *p* > 0.05). Likewise, the EPMT, another anxiety-related behavioral test, revealed no significant differences in time spent in the open arms between the two groups (Figure 1e, *t* = 0.088, df = 17, *p* > 0.05). Working memory was examined using the YMZT. Although the number of entries into each arm was significantly higher in the HFD group (Figure 1f, *t* = 2.189, df = 17, *p* < 0.05), the percentage of correct alterations was the same in both groups (Figure 1g, *t* = 0.076, df = 17, *p* > 0.05). Anhedonia-like behavior was evaluated with the SCT. Sucrose preference among HFD mice was significantly lower than in ND mice (Figure 1h, *t* = 4.539, df = 17, *p* < 0.001). These results suggest that an HFD affects body weight gain, hyperlocomotion, and anhedonia-like behavior.

### 3.2. HFD-Induced Body Weight Gain Does Not Impact Hyperlocomotion or Anhedonia-Like Behavior

First, we examined the association between HFD-induced body weight and behavioral traits, as HFD-induced body weight showed bimodal distribution, with HFD-high responders (body weight greater than 30 g), and HFD-low responders (body weight less than 28 g) (Figure 1a). We, therefore, conducted correlation analysis between body weight and behavioral traits, and found that body weight was not correlated with distance traveled in the OFT (*r* = −0.28, *p* = 0.408, Figure 2a), number of entries in the YMZT (*r* = −0.18, *p* = 0.606, Figure 2b), or sucrose preference in the SCT (*r* = 0.41, *p* = 0.208, Figure 2c).

We further examined the association between HFD-induced body weight gain and behavioral traits, since HFD-induced body weight gain showed mild bimodal distribution, with HFD-high responders (more than 9 g weight gain), and HFD-low responders (less than 8 g weight gain) (Figure 1b). Body weight gain was not correlated with distance traveled in the OFT (*r* = −0.38, *p* = 0.250, Figure 3a), number of entries in the YMZT (*r* = 0.03, *p* = 0.928, Figure 3b), or sucrose preference in the SCT (*r* = 0.24, *p* = 0.480, Figure 3c). These results indicate that HFD-induced hyperlocomotion and anhedonia-like behavior were not attributed to body weight gain in C57BL/6J mice.

### 3.3. A Better Working Memory Correlates with Lower Anxiety in Mice Fed an HFD

Next, we conducted comprehensive histogram analyses using all behavioral traits, and found that the percentage of correct alterations in the YMZT showed a bimodal distribution, with HFD-high responders (more than 70% correct), and HFD-low responders (less than 65% correct), in the HFD group, although HFD did not impact the average percentage of correct alterations (Figure 4a,b, *t* = 4.766, df = 7, *p* < 0.01). The percentage of correct alterations was positively correlated with the time spent in the open arms of the EPMT (*r* = 0.73, *p* = 0.011, Figure 4c) in the HFD group, but not in the ND group (*r* = −0.12, *p* = 0.690, Figure 4d). Time spent in the center zone of the OFT, an anxiety-related behavioral test, was not correlated with the percentage of correct alterations in the YMZT (*r* = 0.03, *p* = 0.922, Figure 4e). These results suggest that an HFD enhances working memory in C57BL/6J mice and reduces anxiety-like behavior in the EPMT but not in the OFT.

## 4. Discussion

In the present study, we found that an HFD significantly affects body weight gain, although we observed a less pronounced effect than that reported in previous studies [19]. We also found that an HFD is associated with increased hyperlocomotion/exploratory activity in the OFT and in the YMZT, as well as with anhedonia-like behavior in the SCT. Surprisingly, HFD-induced body weight gain was not associated with behavioral traits as assessed by the OFT, YMZT, and SCT. However, working memory as assessed by the YMZT was significantly correlated with the amount of time spent in the open arms of the EPMT in the HFD group only, not in the ND group. Several papers have reported that an HFD impacts psychiatric traits and cognitive function in rodents. Consistent with some previous reports [17,18], but not all [19], we observed HFD-induced hyperlocomotion/exploratory activity in the OFT and YMZT, and anhedonia-like behavior in the SCT. In contrast to previous reports [20,21,22], our results indicate that an HFD did not affect anxiety-like behavior in the EPMT or working memory in the YMZT. Possible explanations for this discrepancy include differences in strain and age, including young and adult animals, since ICR mice and aged C57BL/6 mice were used in previous studies [19,20,21,22]. Another possible explanation is the differing feeding periods. For example, in the current study, a 7-week HFD feeding period was employed, which is shorter than the 12 periods reported in previous studies [20]. This suggests that the relatively short-term administration of an HFD is significantly associated with psychiatric behavioral abnormalities but does not impact cognitive impairment.

Since one objective of our research was to examine whether HFD-induced obesity was a risk factor for psychiatric disorders and cognitive impairment, we analyzed the correlation between body weight and behavioral traits, according to a previous paper [35]. We unexpectedly found that HFD-induced body weight was not associated with either hyperlocomotion or anhedonia-like behavior. Likewise, HFD-induced body weight gain was not associated with either hyperlocomotion or anhedonia-like behavior, and HFD-related body weight was not associated with working memory. These results suggest that abnormal psychiatric behaviors and cognitive impairment did not result from HFD-induced obesity in C57BL/6J mice. To investigate this further, we conducted comprehensive correlation analyses between the other behavioral traits found in the HFD group. Intriguingly, working memory as assessed by the YMZT was significantly associated with anxiety-like behavior in the EPMT in the HFD group but not in the ND group. Several studies have revealed a similar behavioral association in adult mice; for example, working memory and sensorimotor gating have been shown to be correlated in inbred C57BL/6NCrl mice [36]. Additionally, an association between repetitive behavior and sociability has been reported in C57BL/6J mice [35]. However, in contrast to previous studies that included ND conditions, our findings are based on HFD conditions and provide the novel insight for male C57BL/6J mice, suggesting that an HFD correlates with enhanced working memory in the YMZT, as well as with reduced anxiety in the EPMT but not in the OFT [37,38,39,40,41,42,43,44]. Although further studies are necessary to examine how an HFD enhances working memory and simultaneously affects anxiety-like behavior, our findings indicate that the relatively short-term application of an HFD might have beneficial effects for memory function.

First, the limitations to our study include our use of CE-2, a grain-based chow diet, for ND, and D12492, a purified ingredient diet, for HFD. The latter differs from the former not only at the fat level but also in terms of fiber type, carbohydrate source, and plant chemicals. Therefore, we could not exclude the possibility that food components other than fat influence behavior. Second, we found that body weight gain with an HFD was not associated with any behavioral traits. Since obesity is characterized by various metabolic abnormalities, such as dyslipidemia, hypertension, and elevated glucose levels [45], these abnormalities might have been affected by the behavioral changes we observed. Third, our findings suggest that a better working memory is associated with reduced anxiety. As this conclusion is based solely on the correlation analysis between the YMZT and the EPMT, further studies that include an examination of working memory and anxiety are necessary to support our findings.

Our results suggest the possibility that an HFD could have beneficial effects on cognitive function and psychiatric traits. Furthermore, although nearly all previous studies have reported that an HFD impairs working memory [20,21,22], our findings suggest that an HFD has a positive effect on working memory in some C57BL/6J mice in which anxiety was reduced. This suggests that an HFD might be beneficial under certain conditions. Correlation analysis among individual traits is a useful tool to determine these limited conditions. Furthermore, determining the associated mechanism by which an HFD elicits response diversity is a critical area for future research. Therefore, we suggest that it is important to determine the conditions under which certain nutrients have positive effects on particular phenotypes.

## 5. Conclusions

An HFD was associated with body weight gain and behavioral abnormalities such as hyperlocomotion and anhedonia-like behavior in male C57BL/6J mice. Contrary to our expectations, body weight gain was not correlated with behavioral abnormalities, suggesting that HFD-induced obesity was not the cause of these abnormalities. Further, an HFD enhanced working memory in male C57BL/6J mice with less anxiety.

## Figures and Tables

**Figure 1 nutrients-12-02036-f001:**
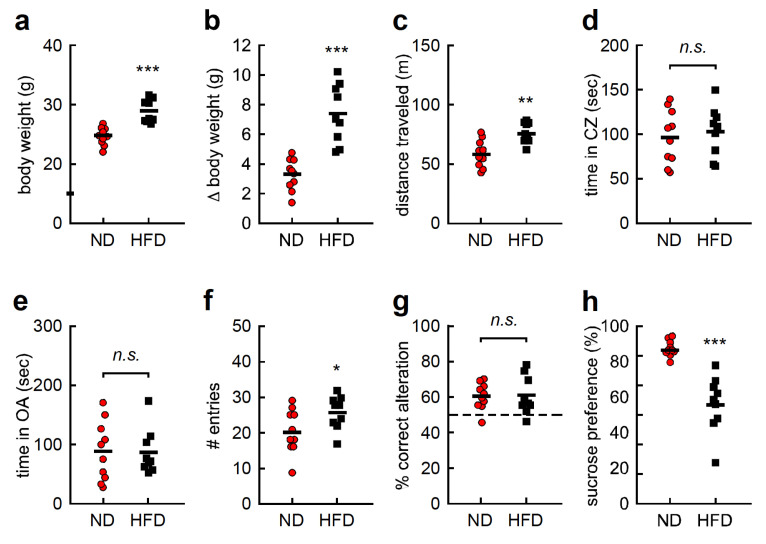
High-fat diet (HFD) affects body weight gain, hyperlocomotion, and anhedonia-like behavior. (**a**) Body weight of each individual. (**b**) Body weight gain of each individual. (**c**) Distance traveled and (**d**) time spent in the center zone (CZ) in the open-field test. (**e**) Time spent in the open arm (OA) of the elevated plus maze test. (**f)** Number (#) of entries into each arm and (**g**) percentage of correct alterations in the Y-maze test. (**h**) Sucrose preference in the sucrose consumption test. Data are presented as dot plot and means as short solid line. OA; open-arm, CZ; center zone, sec; second. * *p* < 0.05, ** *p* < 0.01, *** *p* < 0.001, n.s. not significant.

**Figure 2 nutrients-12-02036-f002:**
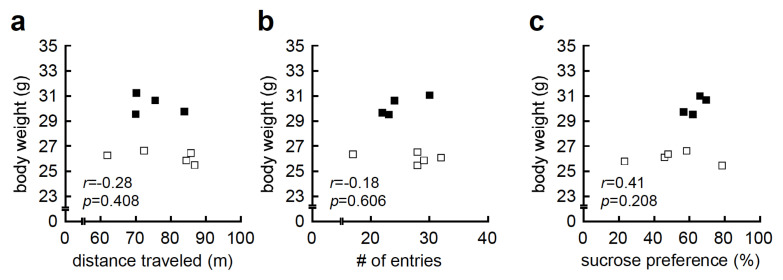
High-fat diet (HFD)-induced body weight gain was not associated with hyperlocomotion or anhedonia-like behavior. (**a**) Correlation analysis between body weight and distance traveled in the open-field test (OFT; *r* = −0.28, *p* = 0.408). (**b**) Correlation analysis between body weight and number (#) of entries in the Y-maze test (YMZT; *r* = −0.18, *p* = 0.606). (**c**) Correlation analysis between body weight and percent sucrose preference in the sucrose consumption test (SCT; *r* = 0.41, *p* = 0.208). Hollow squares indicate HFD-low responders (body weight less than 28 g) and solid squares indicate HFD-high responders (body weight greater than 30 g).

**Figure 3 nutrients-12-02036-f003:**
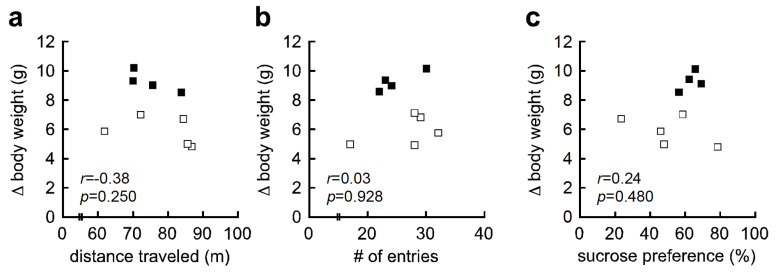
High-fat diet (HFD)-induced body weight gain was not associated with hyperlocomotion or anhedonia-like behavior. (**a**) Correlation analysis between body weight gain and distance traveled in the open-field test (OFT; *r* = −0.38, *p* = 0.250). (**b**) Correlation analysis between body weight and number (#) of entries in the Y-maze test (YMZT; *r* = 0.03, *p* = 0.928). (**c**) Correlation analysis between body weight and percent sucrose preference in the sucrose consumption test (SCT; *r* = 0.24, *p* = 0.480). Δ indicates differential (i.e., body weight differential). Hollow squares indicate HFD-low responders (body weight gain less than 8 g) and solid squares indicate HFD-high responders (body weight gain greater than 9 g).

**Figure 4 nutrients-12-02036-f004:**
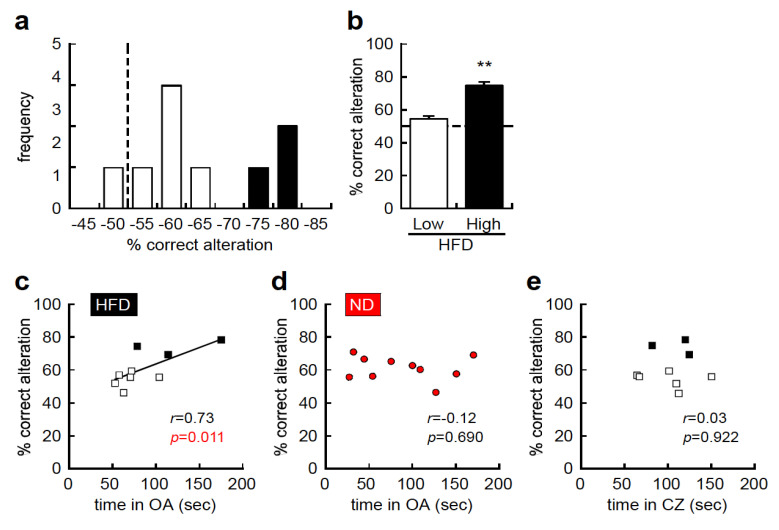
A better working memory correlates with lower anxiety-like behavior in the high-fat diet (HFD) group. (**a**) Histogram representing individual working memory in C57BL/6J mice fed an HFD. The y–axis represents the frequency, and the x–axis represents percentage of correct alterations in the Y-maze test (YMZT). (**b**) Working memory in HFD-low responders (less than 65% correct alterations) and HFD-high responders (more than 70% correct alternations). (**c**) Correlation analysis between working memory and time spent in the open arms (OA) of the elevated plus maze test (EPMT) in mice fed an HFD (*r* = 0.73, *p* = 0.011). (**d**) Correlation analysis between working memory and time spent in the open arms of the EPMT in the ND group (*r* = −0.12, *p* = 0.690). (**e**) Correlation analysis between working memory and time spent in center zone (CZ) in the open-field test (OFT; *r* = 0.03, *p* = 0.922). Dot line indicates expected value in the absence of bias. Hollow squares indicate HFD-low responders (less than 65% correct alterations), solid squares indicate HFD-high responders (more than 70% correct alternations), and red content indicates ND. OA; open-arm, CZ; center zone, sec; second. ** *p* < 0.01.

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
