# Peer review of "High-Fat Diet Enhances Working Memory in the Y-Maze Test in Male C57BL/6J Mice with Less Anxiety in the Elevated Plus Maze Test"

_nutrients, 2020, doi:10.3390/nu12072036_

Round 1

Reviewer 1 Report

The manuscript is much better, but there are a couple of more issues:
They are still showing body weight in 5 different ways (Fig. 1a-c; Fig 2 a and b and also in Fig 3). Not clear why they are still showing body weight in so many different ways. This is un-necessary repetition. Figure 2 and 3 should instead just show the correlations and/or the behavior raw data.

Author Response

Reviewer#1

The manuscript is much better, but there are a couple of more issues:
They are still showing body weight in 5 different ways (Fig. 1a-c; Fig 2 a and b and also in Fig 3). Not clear why they are still showing body weight in so many different ways. This is un-necessary repetition. Figure 2 and 3 should instead just show the correlations and/or the behavior raw data.

We thank you for your comment and we agree with it; the repetition is unnecessary, and it might confuse the readers. Therefore, we have deleted Figures 1b, 2a, 2b, 3a, and 3b, and renumbered the figures in the manuscript. In addition, the main text and figure legends have been revised accordingly (p.3, lines 121, 123, 125, 127, 129, 130, 131; p.4, lines 135, 136, 137, 138, 143, 145, 146, 147; p.5, lines 149, 150, 151, 156, 157, 158, 161, 162, 163).

Reviewer 2 Report

The authors have addressed my comments.

I would ask that the authors remove the figure references and statistics from the discussion, as they are unnecessary. 

Author Response

Reviewer#2

The authors have addressed my comments. I would ask that the authors remove the figure references and statistics from the discussion, as they are unnecessary. 

We thank you for your comment. We have made the necessary changes in the Discussion by avoiding references to figures and deleting statistical data per your suggestion.

This manuscript is a resubmission of an earlier submission. The following is a list of the peer review reports and author responses from that submission.

Round 1

Reviewer 1 Report

Summary:

In this paper the authors look at the behavioral effects of chronic high fat diet using a battery of standard behavioral tests like the open field assay, elevated plus maze, y-maze, and two bottle choice sucrose preference for anhedonia. The authors then perform correlational analyses to attempt to relate HFD effects to aspects of behavior.

Major Points:

- Typos abound. The manuscript would benefit from proofreading throughout by an academic editing service and/or native English speakers.

- It is usually advisable to use a control diet that is sourced from the same materials as a HFD to ensure that the dietary sources are controlled for. In the case of a long-term study like this, it is a significant weakness. The authors should, at a minimum, address this as a limitation.

- I do not see in the figure legends where the size of the groups is reported.

- The positive effect described in Figure 4 is interesting, but the authors are perhaps overselling and overstating it.

Minor Points:

  • Should always try to show individual data points whenever possible. It aids in interpretation.
  • Figure 3 legend should be “HFD-induced body weight gain was not associated with neither hyper-locomotion nor anhedonia-like behavior.”, no?
  • 7 weeks is an interesting, atypical time point to look at chronic HFD effects. Not a dealbreaker though.

Reviewer 2 Report

The aim of the study is to investigate the impact of high fat diet on psychiatric disorders and cognitive impairment.

This is an interesting study, however there are several points that need to be addressed. I have listed them below. 

I do not think that the title of the article is an adequate representation of the study. The authors need to include the word male in the title, as well as revise some of their conclusions. 

The authors use male animals. Can the authors explain why they did not include female animals? How many animals were included in each group? The authors should include this. 

In terms of the diets used the in the study, what were the specifics of each diet? Have these diets been used by others? How was the diet designed? More details are required.

For the weight data, did the authors conduct multiple t-tests? If not, can they include details of the statistics they used?

All t-statstics and df should be included in the results section of the manuscript. 

Did the authors measure metabolites in blood or tissue to confirm that the diet worked?

Did the author measure memory function using another task? If not, I do not think that they can come to the current conclusions. 

I think that the conclusions that the article comes to with weight gain not being associated with behavioral traits is too simplistic and not well investigated.

The manuscript requires significant revisions, they authors need to be more careful with their conclusions and include more data if they have it. More data would be helpful to support their conclusions, but is not a requirement. 

Reviewer 3 Report

The investigators fed 7-week-old mice high-fat diet for 8 weeks and examined behavioral and morphological consequences. The study is interesting and timely, but there are a number of items that need to be addressed. In terms of general comments, there are numerous grammar and spelling errors throughout the manuscript. It needs to be edited by an English-speaking editor. Second, the body weight and body weight gain data are shown several times – in fact in all three figures 1-3. Only one figure is needed to describe the weight gain. Third, the two diets have to be described in more detail. Did the HFD contain hydrogenated fats, saturated fats, or unsaturated fats? Were the two diets of equal caloric value? Did they contain the same amount of carbohydrates? And the amount of food intake also has to be included. Fourth, it is not clear whether they used male or female mice. Male and female mice will respond diametrically different to HFD, so this is important.

Minor questions:

  1. Giving the diet at 7 months of age means that the brain was not fully developed at that age, so this is partially a developmental study and not so much an examination in adult mice. The developmental aspect of the effects should be considered. Hippocampus is not completely developed in mice until 3 months of age. Young versus adult mice may react differently to HFD. This should be discussed.
  2. Figure 1A and Figure 2A appear to show the same thing. One of these should be removed. Not clear how Figure 2A-C improve the study.
  3. The concept of a HFD high responder and low responder is interesting but not at all explained in the manuscript. Not sure this has physiological validity, other than differences between mice in metabolic rates. Was the high- vs. low-responders to HFD related to amount of food intake? This concept has to be described in detail and justified scientifically.
  4. For Figure 4, it is not clear what Figure 4B shows. Is this data for individual mice, only in the HFD group, and what do the X and Y axes represent? This is unclear and needs to be explained further. Overall, each graph has to be explained better in the Figure legends.
  5. The investigators do not need to show both weigh gain and body weight, or – these could all be included in one Figure instead of spread out over 3 figures.
  6. Did the investigators examine both male and female mice?
  7. Line 129: This should read “HFD-induced weight gain did not impact